# The Molecular Detection of Bacterial Infections of Public Health Importance in Hard Tick (*Ixodidae*) Nymphs Collected from the Forest Fringes of Western Ghats in the Goa, Karnataka and Maharashtra States of India

**DOI:** 10.3390/microorganisms12010052

**Published:** 2023-12-28

**Authors:** Gnanasekar Ragini, Hari Kishan Raju, Ranganathan Krishnamoorthi, Ayyanar Elango, Subramanian Muthukumaravel, Ashwani Kumar

**Affiliations:** 1Climate Change, GIS and VBD Stratification/Mapping, ICMR-Vector Control Research Centre, Department of Health Research, Ministry of Health & Family Welfare, GOI, Medical Complex, Indira Nagar, Puducherry 605 006, India; ragisarosekar99@gmail.com (G.R.); krish.1267@gmail.com (R.K.); 2Division of Vector Biology and Control, ICMR-Vector Control Research Centre, Department of Health Research, Ministry of Health & Family Welfare, GOI, Medical Complex, Indira Nagar, Puducherry 605 006, India; elangoar@yahoo.co.in; 3Molecular Epidemiology, ICMR-Vector Control Research Centre, Department of Health Research, Ministry of Health & Family Welfare, GOI, Medical Complex, Indira Nagar, Puducherry 605 006, India; kumaravelmuthuvel@gmail.com; 4Centre for Global Health Research, Saveetha Medical College and Hospital, Saveetha Institute of Medical and Technical Sciences, Saveetha University, Chennai 605102, India; ashwani07@gmail.com

**Keywords:** *Ixodidae*, *Coxiella burnetii*, nymph, *Haemaphysalis spinigera*, flagging, India

## Abstract

A survey was conducted to determine the human tick-borne bacterial infections in the nymphs which were collected from Western Ghats’ fringe forest areas. Tick nymphs were collected using the flagging method from the villages where cases Kyasanur Forest Disease (KFD) were previously reported in the states of Goa, Karnataka and Maharashtra. A total of 200 tick pools consisting of 4587 nymphs were tested by PCR for the detection of bacteria of public health importance, such as *Coxiella burnetii* and *Rickettsia* spp. Of these, four pools (4.8%) in Karnataka and three pools (4.4%) in Maharashtra were positive for *Coxiella burnetii,* while none of the samples from Goa state were positive. *Rickettsia* spp. were positively obtained from Maharashtra (51.5%), Goa (35.42%) and Karnataka (26.19%). The sequence results of *Rickettsia* spp. showed similarity to the spotted fever group *Candidatus Rickettsia shennongii*, *Rickettsia conorii* subsp. *heilongjiangensis* and *Rickettsia* spp. strain *koreansis.* Individuals are entering into the forest areas for various reasons are more likely to infect with *Coxiella burnetii*. and *Rickettsia* spp.

## 1. Introduction

Ticks are haematophagous arachnids that feast on the blood of animals and are found worldwide. They parasitize a wide range of vertebrates and are recognized globally as important vectors for transmitting animal diseases. In fact, they rank as the second largest vector of human diseases after mosquitoes [1]. Hard ticks, specifically those belonging to the *Ixodidae* family, transmit viruses, bacteria and protozoa to both wild and domestic animals. These parasites circulate in an enzootic cycle between animals and ticks. However, when ticks enter the zoonotic cycle, they can also transmit these parasites to humans, who become accidental hosts in this cycle. In India, the study of tick-borne diseases gained significance following the discovery of the transmission of the Kyasanur Forest disease virus (KFDV) in the Shivamogga district of Karnataka State of India in 1957. However, research on other tick-borne bacterial infections has remained limited. In the past decade, several cases of tick-borne diseases have been reported in different states of India. This increase can be attributed to the availability of molecular and immunodiagnostic diagnostic methods to identify infections in humans. Among these infections, *Rickettsia* spp. and *Coxiella burnetii*. are particularly prevalent in humans [2,3]. Tick bites are more commonly encountered by individuals residing in rural areas and those living on the fringes of forests. Infected tick bites in humans often present as acute, nonspecific lesions that can lead to infection. In order to effectively control tick-borne diseases in regions where people live in proximity to forested areas, it is crucial to conduct studies focused on identifying the vectors responsible for bacterial infections and assessing their abundance. The major viral diseases transmitted to humans by ticks in India include Kyasanur forest disease (KFD) [4] and Crimean-Congo haemorrhagic fever (CCHF) [5]. In addition, a case of Ganjam virus was reported in Vellore district of Tamil Nadu [6]. This virus has been isolated from both ticks and humans. Apart from virus transmission, ticks are also known to transmit bacteria to humans, and cases have been reported in several states in India. These include Indian Tick Typhus [7], Q fever [8], and Lyme disease [9], caused by bacteria such as *Rickettsia conorii* subsp. *indica*, *Coxiella burnetii*, and *Borrelia burgdorferi*. Studies on the infections of bacteria on ticks are very few in India. Recently, *Rickettsia conorii* subsp. *raoultii* and *Rickettsia felis* were reported from adult *Haemaphysalis intermedia*, which was collected from Eastern Ghats [10]. In addition, *R. conorii* and *Rickettsia massiliae* were reported from the nymphs of *Rhipicephalus sanguineus* collected from Asian House Shrews in Gorakhpur district, and Indian tick species collected from Asian house shrews [11]. People living in the forest fringes are exposed to tick bites while visiting the forest for their work. Furthermore, studies on bacterial infections on ticks in Western Ghats were available. Therefore, the present study is to investigate the bacterial infections in the nymphs collected from the Western Ghats particularly in villages where previously reported KFD.

## 2. Materials and Methods

### 2.1. Collection of Tick Samples from Forest

Study area: The study area for the present study was the entire Western Ghats region. A total of 48 grids, each covering an area of 75 square kilometres, have been superimposed on the map of this region (Figure 1). Among these grids, seven were randomly selected to cover forest fringes, wildlife sanctuaries (WLS) and plantations in the Western Ghats. From each of these selected grids, four villages were randomly surveyed for ticks using the flag method. The focus of this study was to collect the tick samples from the forest fringe villages located in between two grids falling in Goa, Maharashtra and Northern Karnataka. A total of five man-hours was utilized to collect nymph stage of tick in forest floor by using the flagging method. The flag comprised of a lint cloth (measuring 100 × 70 cm^2^) attached to a wooden stick, which was dragged over an area of approximately 100 square meters at each selected location. The collected nymphs were then transferred to 80% ethanol for preservation. The ticks were identified using standard taxonomical keys [12,13].

### 2.2. DNA Extraction

From each species, 11–25 nymphs were pooled in a 1.5 mL micro centrifuge tube. Ticks were thoroughly washed with sterile ribonuclease-free water prior to DNA extraction. A commercial extraction kit (DNeasy^®^ Blood and Tissue Kit, Qiagen, Hilden, Germany) was used for DNA extraction according to the manufacturer’s instructions. The quality and quantity of the obtained genomic DNA were analysed with a UV-Vis spectrophotometer (Nanodrop 2000/2000, Thermo Fisher Scientific, Waltham, MA, USA).

### 2.3. Molecular Identification of Bacterial Infection by PCR

#### 2.3.1. Detection of *Coxiella burnetii*

The extracted DNA was utilized to amplify Trans1 (5′-TATGTATCCACCGTAGCCAGTC-3′; forward) and Trans2 (5′-CCCAACAACACCTCCTTATTC-3′; reverse) sequences derived from a transposon-like repetitive region of the *Coxiella burnetii* genome with the expected product size of 687 bp as previously reported [14,15]. The PCR was performed in a final reaction volume of 25 μL, using Gotaq polymerase (Promega, Madison, WI, USA) and 1 µL of the DNA sample as template. The amplification process was performed with an Eppendorf thermal cycler. The cycling conditions consisted of an initial denaturation step at 95 °C for 5 min, followed by 35 cycles of denaturation at 95 °C for 30 s, annealing at 56 °C for 1 min, and extension at 72 °C for 1 min. After each cycle, a final elongation step was performed at 72 °C for 10 min. The PCR products were separated by electrophoresis on 1.5% agarose gels, and gel documentation was performed to visualize the results (Figure 2).

#### 2.3.2. Detection of *Rickettsia* spp.

A nested PCR approach was used to amplify a partial segment of the *Rickettsial* outer membrane protein B gene (rompB). In the first round of PCR, the forward outer primer 5′-GTCAGCGTTACTTCTTCGATGC-3′ and the reverse outer primer 5′-CCGTACTCCATCTTAGCATCAG-3′ were used to amplify a 475 bp fragment. A second round of PCR was then performed using the forward inner primer 5′-CCAATGGCAGGACTTAGCTACT-3′ and the reverse inner primer 5′-AGGCTGGCTGATACACGGAGTAA-3′ to amplify a 267 bp product. For the first round of PCR, a final reaction volume of 25 μL was prepared with Gotaq polymerase (Promega), and 1 µL of DNA sample was used as template. Amplification was performed with an Eppendorf thermal cycler. The PCR cycling conditions consisted of an initial denaturation step at 95 °C for 5 min, followed by 35 cycles of denaturation at 95 °C for 30 s, annealing at 56 °C for 1 min, and extension at 72 °C for 1 min. A final elongation step at 72 °C for 10 min concluded each cycle. For the second round of PCR, one µL of the first PCR product was used as template, along with the second round inner forward and reverse primers. The PCR cycling conditions for the second round were the same as those used in the first round. Following amplification, the PCR samples were separated using 1.5% agarose gels, and gel documentation was performed (Figure 3). The amplicon band was cut and purified with NucleoSpin Gel and PCR Clean-up Columns for gel extraction and PCR clean up (MACHEREY-NAGEL). Sequencing was performed using BigDye Terminator Cycle Sequencing (Applied Biosystems, Waltham, MA, USA). The DNA sequences obtained were subjected to BLAST search (http://blast.ncbi.nlm.nih.gov/blast accessed on 11 May 2023) to identify the agent.

### 2.4. Data Analysis

To estimate the probability of infection rate in pooled ticks, a minimum infection rate (MIR) was calculated in Excel with the use of the CDC’s Mosquito Surveillance Software tool, Version 4.0 (https://www.cdc.gov/westnile/resourcepages/mosqSurvSoft.html accessed on 21 May 2023), which calculated the point estimate infection rate and 95% confidence intervals (CI) using pooled data that take into account individual pool samples sizes.

## 3. Results

### 3.1. Molecular Detection of Bacteria in Nymph

A total of 48 pools, consisting of 1082 nymphs, were collected from Goa, while 84 pools with 1946 nymphs from Karnataka, and 68 pools with 1559 nymphs from Maharashtra. These pools were subjected to a molecular PCR test to detect *Rickettsia* spp. and *Coxiella burnetti* infections. Out of these, four pools were positive in Karnataka, with a minimum infection rate (MIR) of 0.21% (95% CI: 0.00–0.41%), and three pools were positive in Maharashtra, with a MIR of 0.19% (95% CI: 0.00–0.41%). The primary vector for KFD, *H. spinigera*, screened for *Coxiella burnetii*, MIR was 0.12% (95% CI: 0.00–0.28%) in Karnataka and 0.11% (95% CI: 0.00–0.32%) in Maharashtra, and none of the species was positive in Goa (Table 1). Sequencing of the amplicons revealed 100% similarity to *Coxiella burnetii*.

In Goa, Karnataka and Maharashtra, positive results for *Rickettsia* spp. infection were found in *H. spinigera*, *Haemaphysalis turturis* and *Haemaphysalis bispinosa*, while *Amblyomma* spp. was positive in Goa and *Haemaphysalis wellingtoni* was positive in Karnataka. Among these, Maharashtra exhibited the highest percentage of positive pools with a MIR of 2.25% (95% CI:1.51–2.98) followed by Goa 1.57% (95% CI: 0.83–2.31) and Karnataka 1.13% (95% CI: 0.66–1.60).

The nymphs of *H. bispinosa* in Goa showed the *Rickettsia* spp. positivity with a MIR of 11.11% (95% CI: 0.00–31.64%), followed by Karnataka 8.70% (95% CI: 0.00–20.21%) and Maharashtra 2.35% (95% CI: 0.74–396%). The secondary vector of KFD, *H. turturis*, showed a higher positivity rate with a MIR of 2.91% (95% CI: 0.40–5.42%) in Karnataka state (Table 2). The positive PCR products were sequenced with a Genetic Analyzer 3130XL from Applied Biosystems, USA. The sequences were compared using NCBI Blast, and a phylogenetic tree constructed using Mega software (MEGA v11.0). The analysis revealed the presence of *Rickettsia* spp. strain *koreansis* and *Candidatus Rickettsia shennongii* in the nymph of *H. spinigera* and *Rickettsia conorii* subsp. *heilongjiangensis* were found in the *H. bispinsona*. The q fever causative pathogen *Coxiella burnetti* was found both in the nymph of *H. spinigera* and *H. bispinosa*.

#### Nucleotide Sequence Accession Numbers

The Gene Bank accession numbers for *Rickettsia* spp. were OR253743, OR468319, OR468320 and OR253744. The Gene Bank accession numbers for *Coxiella burnetti* were OR253742 and OR468321.

## 4. Discussion

Tick-borne diseases are resurging in India. Individuals residing in rural areas, especially those living in villages near forests, are at higher risk of being bitten by ticks. The primary host of these ticks include domestic animals, wild animals, small mammals and reptiles. Larger mammals are often the preferred targets for adult ticks, while smaller mammals are preferred by immature ticks. Immature ticks play a crucial role in transmitting diseases from one animal to another, and even to humans, as they feed on reservoir hosts that carry pathogens. During their life cycle, pathogens acquired at the immature stage can be transmitted to the next stage by transstadial transmission in virus such as KFDV [16]. In some cases, ticks can act as both vectors and reservoirs when pathogens are transmitted transovarially [17]. Transovarial transmission occurred in CCHF virus and bacteria such as *Rickettsia*. These ticks maintain the parasites within their bodies and can transmit diseases while feeding on other hosts.

In total, there are 26 validated *Rickettsia* species that can be divided into four groups: the spotted fever group (SFG), which also includes *Rickettsia conorii*, the typhus group with species such as *Rickettsia prowazekii* and *Rickettsia typhi*, the *Rickettsia bellii* group, and the *Rickettsia canadensis group* [18,19,20]. *Rickettsia* species are obligate intracellular bacteria transmitted primarily through arthropod bites, particularly ticks (Arachnida), fleas (Siphonaptera), and lice (Insecta), which serve as reservoirs and vectors for these pathogens [21]. Of the 26 species and subspecies of *Rickettsia* in the spotted fever group (SFG), 17 have been identified as causing infection in human. In the current study, *Candidatus Rickettsia shennongii* was identified in *H. spinigera* nymphs. This species was previously been reported as the causative agent of spotted fever in *Rhipicephalus haemaphysaloides* ticks in China [22]. *Rickettsia* spp. *strain koreansis* in *H. spinigera* and *Rickettsia conorii* subsp. *heilongjiangensis* in *H. bispinosa* were identified in the nymphs collected in the KFD area. *Rickettsia conorii* subsp. *indica,* the causative organism of Indian Tick Typhus, (ITT) was isolated from *R. sanguineus* ticks collected in India and has never been isolated from patients there [23]. The molecular diagnosis of *Rickettsia* spp. is time consuming and currently multi-locus sequence typing (MLST), targeting the genes 16s rRNA, OmpA, OmpB, and gltA by nested PCR are essential to identify at species level.

The bacterium that causes Q fever was first identified in a 1930 outbreak among abattoir workers. The bacterium was first isolated from a *Dermacentor andersoni* tick [24]. Horizontal transmission of *C. burnetii* from ticks to mammalian hosts has been reported in experimental studies in several tick species, such as *H. bispinosa, Haemaphysalis humerosa,* and *Rh. sanguineus*, through experimental studies [25]. Coxiella-like endosymbionts (CLEs) with genomic similarity of 97% have also been reported in ticks. Our studies have reported for the first time occurrence of *Coxiella burnetii* in the nymphs collected from Kyasanur forest disease (KFD) affected area. Human cases of Q fever have been reported in several states of India [3]. The predominant mode of infection in humans is inhalation, although other modes of infection are not clear [26]. In India, the prevalence of coxiellosis in cattle, determined by PCR, was found to be 24.5% [27]. Human cases of Q fever were reported as early as 1979 in Rajasthan using a complement fixation test [28]. Reports of Q fever in humans and animals have been received from various states of India including Haryana, Punjab, Karnataka, Kerala, Maharashtra, Uttar Pradesh, Orissa and Delhi [5,29]. More recently, human cases have been reported from Puducherry, India, using qPCR methods [30].

In the present study, both *Rickettsia* spp. and *Coxiella burnetii* were identified in *H. spinigera*, the primary vector of KFDV in these areas. *Rickettsia* spp. was also identified in *H. turturis* and *H. wellingtoni*. Since the nymphs are highly anthropophilic in nature, people visiting the forest during the dry month are at higher risk of exposure to them. However, the abundance of these nymphs in the forest were very high during the dry season, particularly from November to April. Control of the nymphs in the forest areas are more challenging as it spread over many places, personal protective measures are to be taken to prevent from tick bites. Immunodiagnostics tools for identifying the bacteria are essential for treatment of these peoples living in Western Ghats areas.

## 5. Conclusions

In the Western Ghats, people living in forest areas or visiting forest in Shivamogga district of Karnataka, Goa and Maharashtra in India are higher risk of exposure to the causative bacteria for spotted fever group and Q fever. During the dry season, the abundance of the nymphal stage is higher in the forest regions. Therefore, diagnosis to detect the bacterial infection is required in the public health laboratory for treat the cases. Future studies are needed to investigate bacterial infection of adult ticks.

## Figures and Tables

**Figure 1 microorganisms-12-00052-f001:**
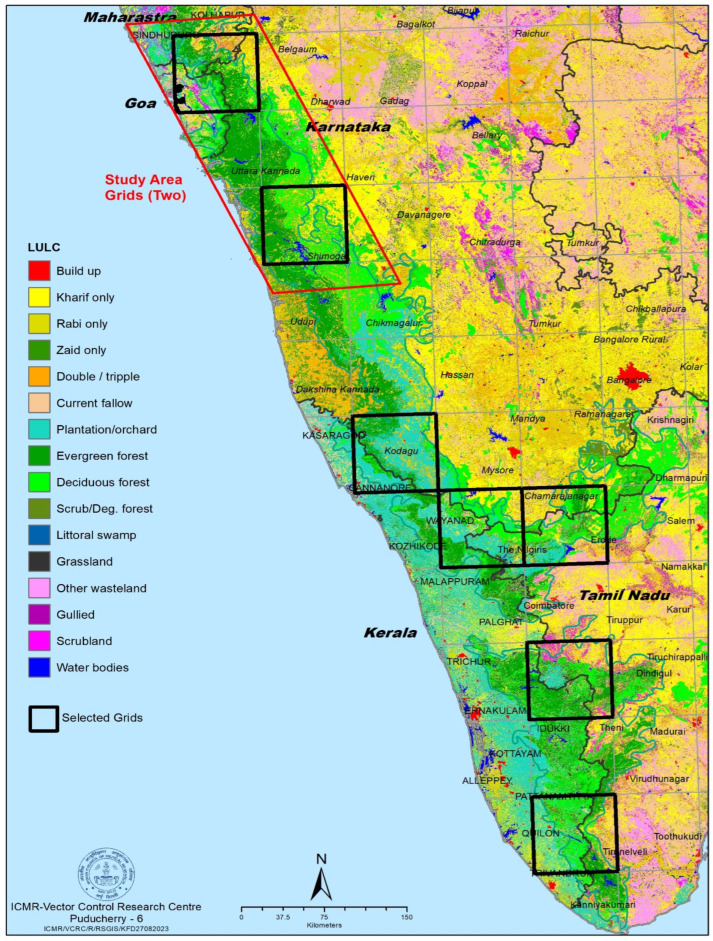
Map showing two grids covering Goa, Maharashtra, and Karnataka states.

**Figure 2 microorganisms-12-00052-f002:**
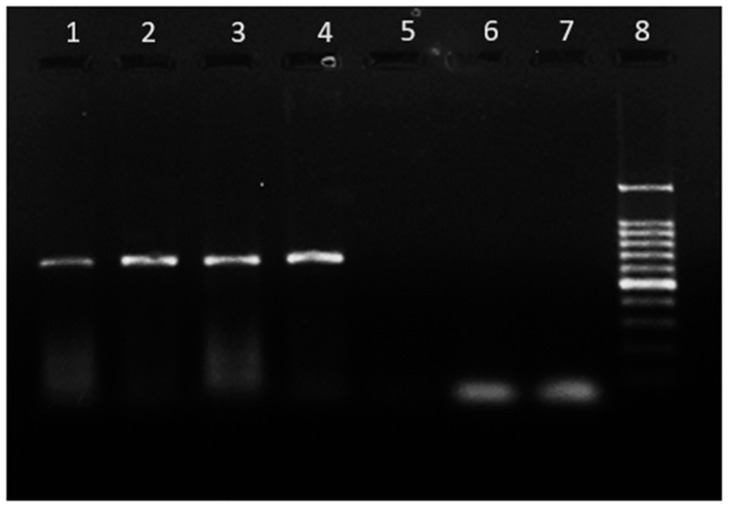
Agarose gel electrophoresis of amplified PCR samples of *Coxiella burnetii*. Legend: Lane 1: *Haemaphysalis spinigera*, Lane 2: *Haemaphysalis bispinosa*, Lane 3: *H. bispinosa,* Lane 4: *H. spinigera*, Lane 5: *Haemaphysalis turturis,* Lane 6: *H. spinigera*, Lane 7: Negative control, Lane 8: DNA size marker 100 bps.

**Figure 3 microorganisms-12-00052-f003:**
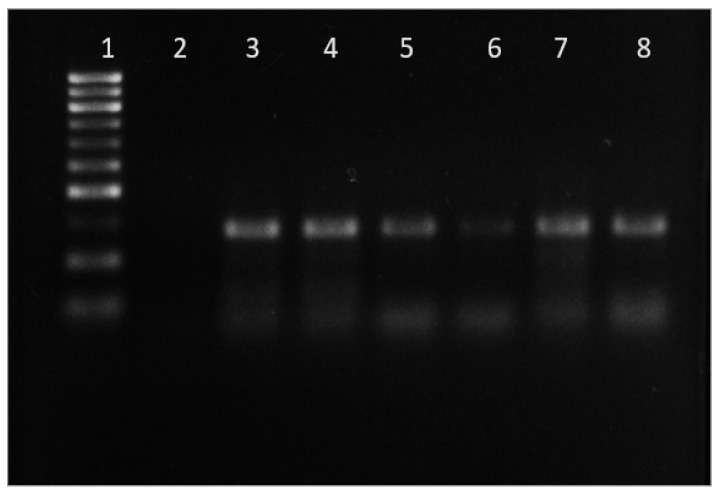
Agarose gel electrophoresis of amplified PCR samples of *Rickettsia* spp. Legend: Lane 1: DNA size marker 100 bps, Lane 2: Negative control, Lane 3: *Haemaphysalis spinigera,* Lane 4: *Haemaphysalis turturis,* Lane 5: *Haemaphysalis bispinosa*, Lane 6: *H. spinigera,* Lane 7: *Haemaphysalis wellingtoni*, Lane 8: *H. bispinosa*.

**Table 1 microorganisms-12-00052-t001:** Summary of *Ixodidae* nymph stage of tick’s positive for *Coxiella burnetii* bacteria.

Tick Species	Goa			Karnataka			Maharashtra		
	No. Pool Tested	No. Pool Positive	Pooled MIR (95% CI)	No. Pool Tested	No. Pool Positive	Pooled MIR (95% CI)	No. Pool Tested	No. Pool Positive	Pooled MIR (95% CI)
*Amblyomma* spp.	1	0	0	0	0	0.0	1	0	0.0
*H. bispinosa*	1	0	0	3	2	8.70 (0.00–20.21)	15	2	0.59 (0.00–1.40)
*H. spinigera*	40	0	0	71	2	0.12 (0.00–0.28)	39	1	0.11 (0.00–0.32)
*H. turturis*	5	0	0	8	0	0.0	11	0	0.0
*H. wellingtoni*	0	0	0	2	0	0.0	0	0	0.0
*Haemaphysalis* spp.	1	0	0	0	0	0.0	2	0	0.0
Total	48	0	0	84	4	0.21 (0.00–0.41)	68	3	0.19 (0.00–0.41)

No.—Number.

**Table 2 microorganisms-12-00052-t002:** Summary of *Ixodidae* nymph stage of ticks positive for *Rickettsia* spp. bacteria.

Tick Species	Goa			Karnataka			Maharashtra		
	No. Pool Tested	No. Pool Positive	Pooled MIR (95% CI)	No. Pool Tested	No. Pool Positive	Pooled MIR (95% CI)	No. Pool Tested	No. Pool Positive	Pooled MIR (95% CI)
*Amblyomma* spp.	1	1	50.00 (0.00–119.30)	0	0	0	1	0	0
*H. bispinosa*	1	1	11.11 (0.00–31.64)	3	2	8.70 (0.00–20.21)	15	8	2.35 (0.74–3.96)
*H. spinigera*	40	13	1.34 (0.61–2.06)	71	14	0.82 (0.39–1.24)	39	21	2.29 (1.32–3.25)
*H. turturis*	5	2	2.20 (0.00–5.21)	8	5	2.91 (0.40–5.42)	11	6	2.45 (0.51–4.38)
*H. wellingtoni*	0	0	0	2	1	2.94 (0.00–8.62)	0	0	0.0
*Haemaphysalis* spp.	1	0	0	0	0	0	2	0	0
Total	48	17	1.57 (0.83–2.31)	84	22	1.13 (0.66–1.60)	68	35	2.25 (1.51–2.98)

No.—Number.

## Data Availability

Data will be available on request.

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
