# Peer review of "The Molecular Detection of Bacterial Infections of Public Health Importance in Hard Tick (Ixodidae) Nymphs Collected from the Forest Fringes of Western Ghats in the Goa, Karnataka and Maharashtra States of India"

_microorganisms, 2023, doi:10.3390/microorganisms12010052_

Round 1

Reviewer 1 Report

Comments and Suggestions for Authors

This manuscript had the aim to investigate the presence of some bacterial pathogens in ticks collected in a geographic area of India. The aim of the study is not clear, because the authors often referred to KFDV (also in the title), even though they did not investigated it.

The authors searched DNA of some bacteria with proper PCR assays, but they often mentioned the bacterial agents in confusing manner. For example, sometimes they referred to Coxiella sp and sometimes to Coxiella burnetii. The used PCR protocol is specific for C. burnetii, not for genus Coxiella.

Discussion and conclusion are not clear because there is the same problem: comment about KFDV, that has not been investigated, and bacterial agents confused.

Author Response

Dear Sir/ Madam

We have modified the manuscript based on the your comments

Response to Reviewer 1

Comments 1: [The accent to KFD should definitely be reduced across the manuscript, and title need to be corrected according to the results present. In my opinion, there no need to mention KFD in the title as far as no KFD study performed.]

Response 1: [Type your response here and mark your revisions in red] Thank you for pointing this out. I/We agree with this comment.

We removed and changed the title as given below

Molecular detection of bacterial infections of public health importance in hard ticks (Ixodidae) nymphs collected from forest fringes of Western Ghats in Goa, Karnataka and Maharashtra states of India”. [We have modified the title in the manuscript. Page No 1, line no 2-5]”

Comments 2: [The authors searched DNA of some bacteria with proper PCR assays, but they often mentioned the bacterial agents in confusing manner. For example, sometimes they referred to Coxiella sp and sometimes to Coxiella burnetii. The used PCR protocol is specific for C. burnetii, not for genus Coxiella]

Response 2: [Type your response here and mark your revisions in red] Thank you for pointing this out. I/We agree with this comment. The used PCR protocol is specific for C. burnetii and we completely changed Coxiella sp to Coxiella burnetii in the manuscript [We have changed from the manuscript Line no 61,69,86,255,328)]”

Comments 3: [Discussion and conclusion are not clear because there is the same problem: comment about KFDV, that has not been investigated, and bacterial agents confused.]

Response 3: [Type your response here and mark your revisions in red] Thank you for pointing this out. I/We agree with this comment. We changed the Discussions and Conclusions.  [We have modified in manuscript Page No. 13, line no 369- 398]”

Thank you for your valuable suggestions 

Reviewer 2 Report

Comments and Suggestions for Authors

Manuscript requires intensive proofreading due to a relatively large number of typos and grammatical errors (e.g. writing spaces before citations, line 134 “ribonuclease water”, it means ribonuclease free?).

In India, the occurrence of bacteria of the genus Borrelia or their vector, ticks of the genus Ixodes, has not been reliably confirmed. To date, there is probably only one publication reporting this occurrence without further confirmation. The absence of their capture therefore has a very limited information benefit. If the authors still wish to present these data, the absence of reliable confirmation of the occurrence of Lyme disease in India should be discussed in more detail.

The contribution of phylogenetic analyzes is also somewhat debatable. I recommend discussing them in more detail or omitting them.

It would be appropriate to describe the sequencing procedure in more detail in the methodology.

Comments on the Quality of English Language

Moderate editing of English language required.

Author Response

Dear Sir/ Madam

We have modified the manuscript based on your comments

Response to Reviewer 2

Comments 1: [Manuscript requires intensive proofreading due to a relatively large number of typos and grammatical errors (e.g. writing spaces before citations, line 134 “ribonuclease water”, it means ribonuclease free?).]

Response 1: [Type your response here and mark your revisions in red] Thank you for pointing this out. I/We agree with this comment. Extremely sorry for that. It is typing error. We have changed into Ribonuclease free water [We modified in the page No. 4, Line no 137)]”

Comments 2: [In India, the occurrence of bacteria of the genus Borrelia or their vector, ticks of the genus Ixodes, has not been reliably confirmed. To date, there is probably only one publication reporting this occurrence without further confirmation. The absence of their capture therefore has a very limited information benefit. If the authors still wish to present these data, the absence of reliable confirmation of the occurrence of Lyme disease in India should be discussed in more detail]

Response 2: [Type your response here and mark your revisions in red] Thank you for pointing this out. I/We agree with this comment. We completely removed the Borrelia part from the manuscript  [We have removed in the manuscript. Page No 4. Line no from 156 to 173 and Page No 6. 238 to 239]”

Comments 3: [The contribution of phylogenetic analyzes is also somewhat debatable. I recommend discussing them in more detail or omitting them]

Response 3: [Type your response here and mark your revisions in red] Thank you for pointing this out. I/We agree with this comment. We have removed phylogenetic tree from the manuscript. [Fig 4 and Fig 5 deleted. Page 9 &10 line number 266 and 284]”

Comments 4: [It would be appropriate to describe the sequencing procedure in more detail in the methodology.]

Response 4: [Type your response here and mark your revisions in red] Thank you for pointing this out.  I/We agree with this comment. We have modified the methodology in the manuscript. The amplicon band was cut and purified with NucleoSpin Gel and PCR Clean‑up Columns for gel extraction and PCR clean up (MACHEREY-NAGEL). Sequencing was done using BigDye Terminator Cycle Sequencing (Applied Biosystems, USA). The DNA sequences obtained were subjected to BLAST search (http://blast.ncbi.nlm.nih.gov/blast) to identify the agent. [modified in the manuscript Page No.5; line 191 to 196]”

Additional clarification

Comments. Moderate editing of English language required.

Response: I/We agree with this comment, accordingly your suggestions we improved the English language by using Grammarly software, if this communication accepted please use MDPI software for editing before final publication.  

Thank you for your valuable suggestions 

Round 2

Reviewer 1 Report

Comments and Suggestions for Authors

The authors revised their manuscript, but further errors arose. In the abstract, it is reported that they searched coxiella, rickettsia and borrelia, but in the text the part about borrelia is not present. Abundant comments on KFDV are still present, but they are not relevant for the purpose of this study. Probably, the aim of the investigation is not clear.

Author Response

Dear Sir/ Madam

We have modified the manuscript based on your comments

Thank you for your valuable suggestions 

Response to Reviewer 1

Comments 1: [In the abstract, it is reported that they searched coxiella, rickettsia and borrelia, but in the text, the part about borrelia is not present.]

Response1: We removed the borrelia part of the manuscript. We removed borrelia part from the abstract.  [We have modified the manuscript. Page No 2, line no 62 & 66]”

Comments 2: [Abundant comments on KFDV are still present, but they are not relevant for the purpose of this study. Probably, the aim of the investigation is not clear.]

Response 2: We have removed the KFDV in many places. We selected the villages for the tick survey based on previous KFD cases reported. We are studying tick prevalence in the KFD-affected areas of Western Ghats. [We have modified the manuscript. Page No 2, line no 57 – 60, page no. 3 line no, 107 – 117, page no. 10 line no. 456 – 460, page no. 11 line 509-514]”
